# Pervasive Sharing of Causal Genetic Risk Factors Contributes to Clinical and Molecular Overlap between Sjögren’s Disease and Systemic Lupus Erythematosus

**DOI:** 10.3390/ijms241914449

**Published:** 2023-09-22

**Authors:** Karen Chau, Yanint Raksadawan, Kristen Allison, John A. Ice, Robert Hal Scofield, Iouri Chepelev, Isaac T. W. Harley

**Affiliations:** 1Division of Rheumatology, Department of Medicine, University of Colorado School of Medicine, Aurora, CO 80045, USA; 2Internal Medicine Residency Program, Louis A. Weiss Memorial Hospital, Chicago, IL 60640, USA; 3Research Service, Oklahoma City US Department of Veterans Affairs Medical Center, Oklahoma City, OK 73104, USA; 4Medicine Service, Oklahoma City US Department of Veterans Affairs Medical Center, Oklahoma City, OK 73104, USA; 5Department of Medicine, University of Oklahoma Health Sciences Center, Oklahoma City, OK 73104, USA; 6Arthritis & Clinical Immunology Program, Oklahoma Medical Research Foundation, Oklahoma City, OK 73104, USA; 7Research Service, Cincinnati US Department of Veterans Affairs Medical Center, Cincinnati, OH 45220, USA; 8Rheumatology Section, Medicine Service, Eastern Colorado Healthcare System, US Department of Veterans Affairs Medical Center, Aurora, CO 80045, USA

**Keywords:** Systemic Lupus Erythematosus (SLE), Sjögren’s Disease (SjD), Sjögrens syndrome, overlap, GWAS, Mendelian randomization, causal gene networks

## Abstract

SjD (Sjögren’s Disease) and SLE (Systemic Lupus Erythematosus) are similar diseases. There is extensive overlap between the two in terms of both clinical features and pathobiologic mechanisms. Shared genetic risk is a potential explanation of this overlap. In this study, we evaluated whether these diseases share causal genetic risk factors. We compared the causal genetic risk for SLE and SjD using three complementary approaches. First, we examined the published GWAS results for these two diseases by analyzing the predicted causal gene protein–protein interaction networks of both diseases. Since this method does not account for overlapping risk intervals, we examined whether such intervals also overlap. Third, we used two-sample Mendelian randomization (two sample MR) using GWAS summary statistics to determine whether risk variants for SLE are causal for SjD and vice versa. We found that both the putative causal genes and the genomic risk intervals for SLE and SjD overlap 28- and 130-times more than expected by chance (*p* < 1.1 × 10^−24^ and *p* < 1.1 × 10^−41^, respectively). Further, two sample MR analysis confirmed that alone or in aggregate, SLE is likely causal for SjD and vice versa. [SjD variants predicting SLE: OR = 2.56; 95% CI (1.98–3.30); *p* < 1.4 × 10^−13^, inverse-variance weighted; SLE variants predicting SjD: OR = 1.36; 95% CI (1.26–1.47); *p* < 1.6 × 10^−11^, inverse-variance weighted]. Notably, some variants have disparate impact in terms of effect size across disease states. Overlapping causal genetic risk factors were found for both diseases using complementary approaches. These observations support the hypothesis that shared genetic factors drive the clinical and pathobiologic overlap between these diseases. Our study has implications for both differential diagnosis and future genetic studies of these two conditions.

## 1. Introduction

The advent of omics-based approaches to disease characterization has prompted the reconsideration of traditional phenotype-based disease classification. SjD (Sjögren’s Disease) and SLE (Systemic Lupus Erythematosus) are two diseases worth considering for reclassification. The observations that these diseases share many features [1,2,3] and overlap substantially [4,5] are not new. A recent focus of the field has been to define a classification criteria that distinguish between these two diseases [6,7]. The goal is to improve the homogeneity of cohorts examined in clinical trials and thereby improve trial success [8]. Nonetheless, several important clinical features overlap. Maternal seropositivity for anti-Ro/SS-A, the characteristic autoantibody for SjD, is the most common cause of neonatal lupus [5]. Many patients share features of both diseases [9], and secondary SjD is found in up to 52% of SLE patients [4,6,10,11]. The clinical diagnostic categories are blurry. Initial diagnosis with another autoimmune disease, such as SLE that is then revised to SjD, is common [12].

In addition to clinical overlap, several pathobiologic mechanisms are shared as well. These shared mechanisms include complement [13], interferon signaling [14,15] and extrafollicular B cell development [16]. Within both diseases, patients fit into stable molecularly defined disease sub-clusters [14,17,18,19,20]. However, a study by the PRECISEADS consortium performed longitudinal multi-omic profiling of persons with diverse systemic autoimmune diseases, including SLE and SjD [21]. Analysis of these data defined four stable clusters of multi-omic signatures for systemic autoimmune diseases that were stable over time regardless of therapy. Importantly, these clusters do not align with traditional disease classification. That is, persons diagnosed with either SjD or SLE were found in similar proportions across all clusters with a preponderance for the interferon signature cluster. Similarly, a persistent anti-Ro (or anti-SSA) + molecular signature of inflammatory gene expression spanning systemic autoimmune diseases has been described [22].

There is also evidence of genetic overlap. Both in family-based studies and nation-wide health records studies, the risk of SLE and SjD in first-degree relatives of patients with either is 5–18× higher [23,24,25]. We previously described rs2431697, a risk variant that disrupts an enhancer for microRNA-146A, as a shared causal risk factor for SLE and SjD [26]. Similarly, SLE and SjD share X chromosome copy number as a major genetic contributor to disease risk [27], and several of the genetic risk loci in SLE and SjD overlap [28,29].

In this paper, we sought to better understand the extent of genetic overlap between these two disease phenotypes. To do so, we collated recent GWAS studies of SjD and SLE. First, we analyzed predicted causal genes and found extensive overlap. Second, we analyzed genomic risk intervals and found extensive overlap. Finally, Mendelian randomization analysis of variants from several published GWAS studies from each phenotype revealed that SLE is likely causal for SjD and vice versa. To our knowledge, we are the first to apply these analytic techniques to evaluate the overlap between SjD and SLE. Our results reveal pervasive sharing of causal genetic risk factors between these two disease phenotypes. They also suggest a revised nosology that either considers SjD as a form fruste of SLE, SLE as a severe form of SjD or that both diseases exist together on a spectrum. This novel paradigm may prove useful as a conceptual approach to these commonly overlapping diseases.

## 2. Results

To address the extent of genetic overlap between SjD and SLE, we applied three methods to compare GWAS results from these diseases. Both detailed description of our methods (putative causal gene network overlap, association interval overlap and Mendelian randomization) and a schematic of our overall analysis are presented in the Section 4.

### 2.1. Sharing of GWAS Identified Genes and Loci between Sjögren’s Disease and Systemic Lupus Erythematosus

#### 2.1.1. SjD Putative Causal Gene Predicted Protein–Protein Interaction Networks

We used a published approach to annotate the predicted causal genes for SjD genetic risk loci from the EBI GWAS catalog [30] (Appendix A). One SjD GWAS study, that of Khatri et al., has not yet been incorporated into the L2G Open Targets Genetics database [31,32]. Therefore, we manually annotated these loci using proxies (Appendix A). We then queried the STRING database [http://string-db.org, accessed on 11 April 2023] using this list to obtain a predicted protein–protein interaction network for SjD (Figure 1 and Appendix A).

Like the published causal gene networks for monogenic and polygenic SLE [33,34,35] (Table 1, Appendix A), the SjD genes are predicted to interact more significantly than would be expected by chance (*p* < 5.5 × 10^−16^). The SjD genes also show a relatively high degree of clustering. Taken together, this suggests that these genes regulate the common pathways within a shared molecular network. This finding also supports the annotation of the individual genes as putatively causal.

#### 2.1.2. Pathway Analysis of SjD Putative Causal Genes

We then performed pathway analysis to further characterize the list of putatively causal genes responsible for SjD development. Links to pathway analyses can be found in (Appendix A). These analyses further validated the contribution of this gene network to known SjD pathobiology. Specifically, these analyses found pathway enrichment for IFN signaling (both type I and interferon-gamma), transcription by NF-kB and autoimmune diseases where secondary SjD or overlap is common (systemic sclerosis, SLE, rheumatoid arthritis and primary biliary cholangitis/primary biliary cirrhosis). One finding of particular interest is the overrepresentation of the IL-35 signaling pathway in this network. This pathway has been found to be dysregulated in several studies of patients with SjD [36,37,38].

#### 2.1.3. Extensive Overlap between SLE and SjD Putative Causal Gene Networks

To determine the extent of overlap between our SjD risk network and SLE risk genes, we compared the overlap between putative causal genes for SLE [33,34,35] (Appendix A) and SjD (Appendix A). This analysis revealed extensive overlap between the predicted causal genes for these two diseases (Figure 2).

Indeed, this overlap was much greater than expected by chance (Table 2).

#### 2.1.4. Extensive Overlap between SLE and SjD GWAS Loci

To address the limitations of the putative causal gene network approach (see Section 3), we also calculated the overlap between GWAS loci. We defined a locus as the 500,000 bp window centered on a lead variant from the EBI/NHGRI GWAS catalog, similar to a published approach [39]. We found extensive overlap of SjD GWAS loci with SLE GWAS loci (Table 2 and Appendix A). However, examining overlapping loci results in a more moderate estimate of enrichment over expectation as compared to putative causal genes. This may be due to, in part, the way we define regions. This definition results in four SJD-associated regions within the extended MHC locus, of which none are within 500,000 bp of the MHC signal for SLE (Appendix A). A prior study indicates that the major association of MHC with SLE and SjD is the same and is driven by alleles determining the complement factor C4 copy number [13].

In sum, these analyses revealed extensive overlap at both the causal gene and locus level between SjD and SLE.

### 2.2. Mendelian Randomization Analysis of Variants Associated with Sjögren’s Disease and Systemic Lupus Erythematosus

The causal gene overlap, but interval overlap approaches do not necessarily imply sharing of risk alleles. Thus, we also applied an analysis technique that evaluates causal relationship between phenotypes with respect to shared risk alleles, Mendelian randomization (MR).

#### 2.2.1. Analysis 1: The Causal Effect of SjD Variants on SLE in European Ancestry GWAS

We started with lists of genome-wide significant (*p* < 5.0 × 10^−8^) SjD risk variants from European ancestry GWAS studies (Appendix A). We used the ‘TwoSampleMR’ R package to analyze the effect of these variants in the summary statistics of a publicly available SLE GWAS in a cohort of persons with European ancestry [40]. Inverse-variance weighted (IVW) MR analysis was the primary inference. We supplemented this with weighted median, weighted mode, and MR Egger analyses. These additional analyses were used to evaluate the robustness of the IVW results. These analyses revealed evidence for a causal relationship between these two phenotypes mediated by these variants or linked variants. Importantly, each of the four MR methods broadly agreed in terms of direction and magnitude of effect size (Table 3). This agreement indicates a stable and robust causal relationship between SjD and SLE in this population.

The forest plot of effect sizes (Figure 3A) shows this consistent effect of SjD on SLE estimated using these risk variants. To ensure that the causal relationship between SjD and SLE was not overly influenced by any single variant, we performed the leave-one-out sensitivity analysis. This analysis did not show that removal of a single variant markedly changes the effect size estimate (Appendix A). In the scatter plot of SjD and SLE effect estimates (Figure 3B), the slope of the regression line for each method indicates a consistent estimate of overall effect size across methods (Figure 3B). The plot also revealed an MR Egger regression intercept of −0.072 that does not significantly differ from zero (Figure 3B and Appendix A). This finding is consistent with the absence of directional pleiotropy [41] and the InSIDE assumption. Taken together, there is a strong dose-dependent relationship between the genetic associations for SjD and SLE.

A funnel plot did not reveal marked asymmetry. This is consistent with the absence of effect heterogeneity [42]. However, despite the consistency in estimated MR effect size across variants and methods, there is evidence of significant effect heterogeneity for both IVW and MR Egger analyses (Appendix A and Appendix A). We suspect that this heterogeneity arises from the broad range of estimated odds ratios for the variants selected for both SjD [1.13–2.08] and SLE [1.05–2.48]. Additional results and sensitivity analyses are presented in (Appendix A, and Appendix A).

Taken together, these analyses indicate a causal relationship between SjD and SLE in these cohorts of persons with European ancestry.

#### 2.2.2. Analysis 2: The Causal Effect of SLE Variants on SjD in European Ancestry GWAS

We next performed the reciprocal analysis by using SLE associated variants to evaluate the causal relationship between SLE and SjD. To do this, we used the genome-wide significant (*p* < 5.0 × 10^−8^) SLE risk variants from a large GWAS of SLE in a cohort of persons with European ancestry [40] (see Section 4 for details) (Appendix A). For the outcome phenotype, we used the GWAS summary statistics found by searching for “sjogren syndrome” in the MR-Base platform [43]. These summary statistics are derived from the FinnGen Biobank GWAS (M13_Sjogrens) [44]. MR analyses were carried out as in Section 2.2.3. These analyses revealed evidence for a causal relationship between SjD and SLE mediated by SLE variants as well. Importantly, each of the four MR methods broadly agreed in terms of direction and magnitude of effect size (Table 4). This agreement indicates a stable and robust causal relationship between SjD and SLE in this population.

The forest plot of MR effect sizes (Figure 4A) shows a more moderate effect of SLE variants on SjD. This is consistent with the overall estimates (Table 4). This contrasts with the estimated effect of SjD on SLE (Table 3). Notably, the Finnish population is something of an outlier in comparison to other European populations [45]. This difference in effect sizes may have to do with different LD patterns across populations. It may also be that the SLE on SjD effects are, in fact, less than the effects of SjD on SLE. There was also evidence of effect heterogeneity for both IVW and MR Egger analyses, but less significant than in the reciprocal analysis (Section 2.2.3).

We performed sensitivity analysis as in the reciprocal analysis (Section 2.2.3). The leave-one-out sensitivity analysis does not show that any single variant markedly changes the overall effect size estimate (Appendix A). Like the previous analysis, in the scatter plot of SjD and SLE effect estimates, the slope of the regression line for each method indicates a consistent estimate of overall effect size across methods (Figure 4B). This analysis also revealed an MR Egger regression intercept of −0.089. This intercept was of a similar size to that seen in the reciprocal analysis (Section 2.2.3). Due to the larger number of SNPs used in this analysis, the standard error of this intercept is smaller, leading to a nominally significant *p*-value for directional pleiotropy (Appendix A). This finding is consistent with the presence of directional pleiotropy or violation of the Instrument Strength Independent of Direct Effect (InSIDE) assumption [41]. However, the deviation from zero is small in absolute terms. Despite this, the MR Egger-derived effect estimate is significantly positive. While there is evidence of directional pleiotropy, once it is accounted for, a significant causal relationship remains. Taken together, there is a strong dose-dependent relationship that holds for most of the genetic associations for SLE and SjD.

One variant merits separate consideration. The SNP, rs150180633, sticks out in the scatter plot as it does not follow the rest of the variants. The effect direction comparing SjD and SLE is opposite, and the standard error is relatively large (Figure 4B). This variant represents a distant secondary effect in the extended HLA region from the GWAS of SLE performed in a cohort of persons with European ancestry. It is nearly 1 megabase away from the main HLA-region effect, rs389884 [46]. Thus, due to this long distance, this variant may be particularly susceptible to differences in LD pattern between populations. Importantly, this variant is also uncommon in the Finnish population (minor allele frequency, MAF < 1%). Additional supportive data and sensitivity analyses are presented in (Appendix A, and Appendix A). Consistent with the foregoing discussion, removal of this variant led to larger causal effect estimates of SLE for SjD, more significant association and decreased heterogeneity.

Taken together, these results indicate a causal relationship between SjD and SLE in these cohorts of persons with European ancestry.

#### 2.2.3. Analysis 3: The Causal Effect of SjD Variants on SLE in East Asian Ancestry GWAS

The results in Section 2.2.3 and Section 2.2.4 point toward a causal relationship between SjD and SLE mediated in European populations. The SjD GWAS studies published to date in East Asian ancestry population have smaller sample sizes than European ancestry SjD GWAS studies. As a result, the associated variants available for MR analysis are fewer and have less precise estimates of effect size (Appendix A). In other words, for any given genetic association, smaller sample size results in a larger standard error. In addition to smaller sample size, the full, genome-wide summary statistics from East Asian ancestry GWAS of SjD have not been made publicly available. This precludes examination of SLE effects in SjD GWAS studies of persons with East Asian ancestry. Nonetheless, we wondered whether the causal relationship between SjD and SLE that we identified could be independently validated. We started with the list of genome-wide significant (*p* < 5.0 × 10^−8^) SjD risk variants reported within the manuscripts reporting several East Asian ancestry SjD GWAS studies (Appendix A). We again used the ‘TwoSampleMR’ R package to analyze the effect of SjD on SLE using the summary statistics of a large, publicly available SLE GWAS in a cohort of persons with East Asian ancestry [47]. MR analysis parameters were the same as above. The analyses revealed inconsistent evidence for a causal relationship between these two phenotypes estimated by these variants. Importantly, IVW and the outlier selection and removal-based methods (weighted median and weighted mode) agreed broadly in terms of direction and magnitude of effect size (Table 5). However, MR Egger did not agree with the other methods. MR Egger resulted in a highly imprecise estimate of the MR effect size that could not be distinguished from the null (Figure 5A). Therefore, this result was not significant. The purpose of MR Egger is to serve as a kind of sensitivity analysis that was developed to model directional pleiotropy [48]. It applies Egger regression to the estimation of the MR effect size. Egger regression was originally developed to detect small study bias in meta-analysis. In the MR scenario, it detects bias toward either high or low values from weaker genetic variants. To do so, it performs regression analysis of the variant effects between the exposure and outcome phenotypes. In our case, this is depicted in the scatter plots (Figure 5B). Thus, an MR Egger intercept term that significantly differs from 0 provides evidence of directional pleiotropy. Like the non-significant MR Egger estimate (Table 5), the MR Egger intercept could not be distinguished from 0, as the standard error was larger than the estimate by a considerable margin (Appendix A). Visual inspection of the scatterplot provides an explanation for this. First, there are few variants (*n* = 4) to fit a regression line, as evident from the results of few SjD GWAS studies having been performed with a large sample size. Thus, the regression is more likely to be influenced by outliers or extreme values. Second, outliers are present: both the weighted median and weighted mode MR methods provide lower estimates of MR effect size than IVW. Therefore, the IVW estimate is being biased upwards by outliers. MR Egger tests for a dose–response relationship between exposure effect size and outcome effect size. The precision of the MR Egger estimate relies both on the proportion of variance in the exposure explained by the genetic variants and on the variability between genetic associations with both the exposure and the outcome [41]. For these four variants, the proportion of the variance of SjD explained by each is small since they are the first loci identified as risk loci in cohorts of persons with East Asian ancestry. Likewise, there is considerable variability in these associations both with the exposure and the outcome. Thus, both the estimated MR effect and the estimated intercept produced via MR Egger for these variants are highly imprecise. Thus, the MR Egger analysis indicates either the presence of pleiotropy or a causal effect. Together with the consistent estimates of a causal effect via the IVW, weighted median and weighted mode methods, a causal effect seems more likely. Additional supportive data and sensitivity analyses are presented in (Appendix A, and Appendix A).

Taken together, our analyses suggest a causal relationship between SjD and SLE in these cohorts of persons with East Asian ancestry. However, a scenario where each trait is independently influenced by the same genetic factor cannot be excluded. This scenario is termed as horizontal pleiotropy.

#### 2.2.4. Analysis 4: The Causal Effect of SLE Variants from East Asian Ancestry GWAS on SjD in European Ancestry GWAS

Next, we explored whether the causal relationship we found between SLE and SjD in European ancestry populations was more generalizable. To do so, we evaluated the causal relationship between SLE and SjD using GWAS of SLE in persons with East Asian ancestry [47] and a GWAS of SjD in persons of European ancestry [44]. We again restricted the variants in question to those that passed the genome-wide significance (*p* < 5.0 × 10^−8^) in the exposure GWAS of SLE. We also applied clumping and pruning to remove LD between variants using the ‘TwoSampleMR’ R package. Analysis parameters were the same as the other analyses. This analysis revealed consistent evidence for a causal relationship between these two phenotypes mediated by these variants (Table 6 and Figure 6). In single variant analysis, several of the SLE risk variants (rs16870693, rs3800387, rs55701306 and rs16869875) show consistent evidence against a causal relationship between SLE and SjD (Appendix A). Notably, each of these variants is uncommon in the European ancestry population examined. The minor allele frequencies (MAFs) are all less than 3.5%. In contrast, the MAF for these variants in East Asian ancestry samples (EAS) from the 1000 Genomes Project ranges from 11.41 to 42.76% [49,50]. This suggests that these variants may be ill-suited to detect casual relationships across populations. That is, they may be particularly susceptible to differences in local LD structure varying across populations. As a result, the relevance condition for instrumental variables may not hold. Despite the challenge with comparing causal effects across populations, the general agreement of the four analyses presented in this manuscript argues for a stable and robust causal relationship between SLE and SjD. Additional supportive data and sensitivity analyses are presented in (Appendix A, and Appendix A).

Taken together, our analyses suggest a causal relationship between SLE in persons with East Asian ancestry and SjD in persons with European ancestry.

## 3. Discussion

The causal gene network and gene interval results provide strong support of a causal relationship between SjD and SLE mediated by shared genetic risk factors. Prior studies investigating the clinical, serologic, immunologic, and molecular overlap between these two conditions also support this hypothesis. Thus, our conclusion may not be surprising. However, the degree of overlap perhaps is. Only three predicted causal genes do not overlap between SLE and SjD (Figure 2). One of the genes, *TNPO3,* is adjacent to the shared gene, *IRF5*. While this region is shared, this gene was not algorithmically assigned as a predicted causal gene for SLE. Notably, a prior study indicated that there are multiple independent genetic associations with SLE in this region [51]. The other two genes that are not shared reside in the same SjD-associated locus on chromosome 17. We did include the variants at this locus in the SjD for SLE MR analysis. However, the variants at this locus were excluded by the algorithm in the ‘TwoSampleMR’ R package from the final analysis due to allele harmonization issues. Full analysis parameters and logs can be found at the link in the Data Availability section. This SjD-associated locus has not been reported as being associated with SLE and encodes the genes *RPTOR* and *CHMP6*. Understanding how this risk locus confers risk for SjD will be important to comprehend how the etiology of SjD is distinct from that of SLE.

The Mendelian randomization results also support a strong causal link between these two disease phenotypes mediated by causal genetic variants. Two observations strengthen this causal link. First, we observed that this relationship is reciprocal: SjD is causal for SLE and vice versa. Second, this relationship is evident in both European and East Asian ancestry populations with caveats (discussed below). Despite these interpretive caveats, the causal link between SjD and SLE revealed via MR is both stable and robust.

One general limitation of our study is that the common genetic contribution to SLE has been more extensively studied than that of SjD. Thus, there are more putative causal genes, associated intervals, and variants for SLE than for SjD. This could cause us to have an under or overestimation of the true extent of overlap between SLE and SjD, depending on what future GWAS of SjD reveal. In addition, there are limitations of the putative causal gene and gene interval approaches. First, two drawbacks of the putative causal gene network approach should be noted. One is that the causal gene at many risk loci cannot be confidently assigned. If these loci are shared, this could cause the overlapping causal gene network to underestimate the true extent of overlap. Another is that as the number of potential overlaps in a population gets larger, so does the number of expected overlaps. Applied to our situation, since 55 SLE risk loci cannot be confidently assigned a putative causal gene, a gene-based approach likely over-estimates the true extent of overlap. As for limitations of MR, one limitation is that there are only five genetic risk variants identified to date for SjD in East Asian ancestry populations. Of these, four remained after LD pruning (Table 5 and Figure 5). As a result, MR Egger cannot distinguish between true causal association and directional pleiotropy. The second caveat is that we have not been able to test whether SLE is directly causal for SjD in GWAS from East Asian ancestry populations. This is because genome-wide summary statistics for SjD GWAS studies in East Asian ancestry cohorts are not available within the MR-Base repository. Instead, we tested whether SLE risk variants identified in a cohort of persons with East Asian ancestry were causally linked to SjD in a cohort of persons with European (Finnish) ancestry (Figure 6). Despite the application of MR across ancestries, our results generally reinforce MR analysis of SjD and SLE in European ancestry populations. Importantly, our analyses of causal links between SjD and SLE used complementary approaches (overlap of putative causal gene networks, intervals and MR analysis), each of which addressed the limitations of the others. Also important, they all led to the same broad conclusion: genetic risk factors are extensively shared between SjD and SLE.

There are several interesting questions regarding the causal relationship between SjD and SLE. Are the effect sizes the same? The MR analyses generate a wide range of causal estimates. Most fell within the range [0.23–1.18]. In both analyses evaluating the impact of SLE variants on SjD risk, the estimated effect sizes in SjD were lower [0.23–0.55] (Table 4 and Table 6). These analyses suggest that the causal effect of SjD variants on SLE is larger. However, several points should be considered in their interpretation. First, the number of GWAS-identified variants is smaller for SjD as compared to SLE. Second, in both instances the comparison GWAS for SjD used to determine the effect of SLE on SjD is derived from the FinnGen study. Amongst European ancestry populations, the Finnish population does show some inter-population differences [45]. Third, the FinnGen study ascertains disease status based on electronic health record (EHR) diagnosis codes. The cohort of participants selected via these codes could include persons who do not, in fact, meet classification criteria for SjD. If so, we would expect smaller effect size estimates for SjD in the FinnGen population relative to a cohort composed exclusively of SjD patients who meet the classification criteria. Other potential explanations include the impact of factors (genetic, environmental, or stochastic immune response), or the so-called “Winner’s Curse”. That is, by nature of their discovery, disease-associated variants tend to systematically overestimate the effect size on a given phenotype relative to the true effect size in a population [52,53]. Thus, several factors may contribute to a downward bias of SjD effect size estimates. Future investigation will be necessary to resolve these possibilities.

Another question is related to the portion of genetic risk for these diseases that is shared. Is only a core network shared between SjD and SLE or is there a broader sharing of causal risk variants? The genomic risk variant footprint for most common human genetic phenotypes comprises two parts. The first part is composed of risk variants identified in initial GWAS scans. These variants constitute a core network of disease risk alleles. The second part is composed of the aggregate effects of thousands to millions of risk alleles spanning the entire genome. Each of these variants add or subtract a small amount of risk [54]. In some instances, gene–gene interactions will likely synergistically contribute to this risk. Our study, by necessity, can only evaluate GWAS studies where there is publicly available data. For SjD, this results in a relatively small number of risk loci. Thus, we may have only addressed the core genetic risk network for SjD. Two methods that may be able to address this, at least in part, are genetic correlation and polygenic risk score calculation. These methods do not only evaluate the effect of the most strongly associated variants but can also include the contribution or risk variants across the entire genome. The SjD GWAS by Khatri et al. [29] did compare genetic correlation, r_g_, between GWAS of SLE and SjD in cohorts of persons with European ancestry. This metric calculates the correlation between effect size of variants spanning the entire genome and not just those that are strongly associated with disease. Their analysis found a high degree of genomic correlation between GWAS for these two phenotypes as compared to other disease phenotypes. Likewise, some approaches to polygenic risk scores (PRS) calculation use the entire genome or include a broader subset of weakly associated (i.e., *p* < 1.0 × 10^−5^) variants. Calculating cross-disease PRS would be another approach to address shared causality between phenotypes. There are tools that allow for the derivation of PRS from summary statistics [55]. However, to our knowledge, current tools do not allow for the estimation of individual-level PRS based on summary statistics. This question warrants further study with individual level genotype data.

A related question also arises from the above consideration: is there a subtype or subtypes of SjD and SLE that exhibit a greater degree of sharing? For example, do anti-Ro/SS-A+ or anti-La/SS-B+ SLE patients exhibit a greater degree of shared genetic risk to SjD patients that exhibit these seropositivities? It may be that loss of immune tolerance to self within these autoantibody systems is under particular genetic control. Indeed, this appears to be the case for anti-Sm autoimmunity, which is restrained by CD72 under normal circumstances [56]. If this is generally true, then it may be that sharing of genetic risk between SjD and SLE is more pronounced in genes that also affect this shared pathway. Studies examining this question will require GWAS of large sample sizes to compensate for the loss of power with substratification of these phenotypes. A related question is how do other phenotypes associated with SjD compare in terms of causal risk variant sharing? Secondary SjD is found with several other autoimmune disease phenotypes, including primary biliary cholangitis, rheumatoid arthritis, systemic sclerosis, and autoimmune thyroid disease. Future studies examining the causal links between SjD and these diseases will be of interest.

The proponents of precision or personalized medicine envision a world where treatment of disease is personalized to the characteristic features of an individual person’s disease process. The hope is that with knowledge of the human genome and the advent of other omics technologies, healthcare practitioners will be able to precisely stratify diseases to smaller, molecularly defined subsets. Of course, delivering the correct therapy to the correct patient at the correct time is a worthy improvement goal for all of healthcare. However, a reductionistic focus on precision, allowed by increasingly precise technical advancements could carry costs in terms of accuracy of disease categorization. The history of revising disease understanding when old paradigms were overturned by new technologies is instructive [57]. Whereas dropsy was once considered a single disease process, most medical students can facilely review a differential diagnosis of common causes for this condition (heart, kidneys, liver, etc.). This understanding of dropsy, comprising distinct entities, is rooted in a mechanistic understanding of how hemodynamics and protein production impact oncotic pressure [58,59]. Similarly, advances leading to the germ theory of disease and molecular genetics led to the reclassification of diseases. For example, diverse clinical presentations including variable presentation of a honey-colored skin rash, inflammatory arthritis, frothy urine, sore throat and chorea could be reclassified as being related to group A streptococcal infection [60]. In the same vein, Mendelian genetic syndromes could be reclassified as related based on action within a defined biochemical pathway. At first glance, abnormally dark urine, liver failure, relative absence of skin pigmentation and intellectual disability are not necessarily related symptoms. However, each of these corresponds to a genetic syndrome caused by loss of function in phenylalanine metabolism (homogentisic acid oxidase, fumarylacetoacetate hydrolase, tyrosinase, and phenylalanine hydroxylase, respectively) [61,62,63,64]. Thus, prior paradigm shifts leading to the reclassification of disease entities improved our understanding and treatment of disease in both directions. Lumping similar entities based on shared mechanisms and splitting based on new mechanism-defined subphenotypes both led to improved disease understanding.

Our findings are consistent with a developing notion that targeting common pathways shared across autoimmune diseases may lead to improved therapeutic responses. One example is the recent application of low-dose IL-2 to expand T regulatory cells across multiple autoimmune and inflammatory disease states [65]. This trial of persons diagnosed with 1 of 11 autoimmune was geared towards demonstrating the safety of this approach in terms of effector T cell activation. However, these data also demonstrated significant improvement in aggregate measures of disease activity from baseline. There is an increasingly precise understanding of the shared pathobiology underlying the clinical overlap between SjD and SLE. Accurately understanding the shared pathobiology of these diseases presents an opportunity to advance therapeutically relevant understanding of both. Since SjD and SLE are diseases with polygenic contributions to disease risk, we wondered what degree of shared genetic risk might be present.

Finally, our findings have implications for future studies of SLE and SjD. They suggest that application of meta-analysis methods to jointly model shared phenotypes may be a way to increase power to detect genetic variants in both SjD and SLE [66,67,68,69]. Combining summary statistics from individual disease GWAS or using individual level genotype data to substratify individuals based on shared phenotype (i.e., combining persons with primary SjD together with persons with SLE and secondary SjD) will increase the effective sample size. In this way, discovery of genetic risk factors for SjD may be accelerated. There are also potential implications impacting clinical care. Most trials in SjD (and SLE for that matter) have not met their primary endpoint. Indeed, it is noteworthy when a trial of therapy for SjD does meet its primary endpoint [70,71,72]. As we have noted, there is clinical and pathobiologic similarity between SjD and other diseases, like SLE and RA. Similarly, some persons are affected by SjD and another autoimmune disease, like SLE or RA (secondary Sjögren’s Disease). Because of this overlap, clinical treatment guidelines of constitutional and non-sicca organ-based manifestations of SjD rely on extrapolation from diseases like SLE and RA [73,74]. Our finding that SLE and SjD are causally linked add credence to this clinical practice. Future studies to better define shared pathways between SjD and other diseases outlined above may further inform therapeutic selection for these manifestations in SjD.

## 4. Materials and Methods

### 4.1. Overview of Analysis Scheme

A schematic of our analysis is presented below (Figure 7).

### 4.2. Determining the Overlap of GWAS-Associated Loci in SLE and SjD

#### 4.2.1. Defining SLE and SjD Loci

We applied a previously described approach [30,75] to define disease-associated genetic loci that have been identified in published genome-wide genetic association scan (GWAS) with Sjögren’s Disease. To do this, we first searched the EBI/NHGRI GWAS catalog [76] using search term “Sjogren syndrome” for all variants associated with Sjögren’s syndrome *p* < 5.0 × 10^−8^. We also included a novel loci from a recently published GWAS of SjD [29] that has not yet been incorporated into the GWAS catalog. A locus was defined as any sequence of adjacent associated variants within 250,000 base pairs. We used the list of SLE-associated loci from a previously published paper that was derived using the same method [30]. The SjD and corresponding SjD association intervals are listed in (Appendix A).

#### 4.2.2. Pathway Analysis of Putatively Causal SjD Risk Genes

We performed pathway analysis on the list of putatively causal genes for SjD using the STRING database [77], Enrichr [78], ToppGene [79] and the Network Data Exchange at NDEXBio.org (accessed on 6 June 2023) gene set enrichment analysis tools [80]. The links to these queries can be found in (Appendix A).

#### 4.2.3. Calculating Locus-Level Overrepresentation between SLE and SjD

Using the two lists of disease-associated loci defined in this way, we then calculated whether the overlaps between these sets of loci in terms of genomic interval was greater than expected by chance. To do this, we wrote a python script that counts the number of overlapping loci and between these diseases (https://github.com/harleyi/sjd_sle_overlap, accessed on 24 May 2023). We then used these overlaps to calculate the fold overrepresentation between these two lists across the genome. The total number of base pairs in the genome was derived from the genome reference consortium, Human Genome Assembly GRCh38.p14 (https://www.ncbi.nlm.nih.gov/grc/human/data, accessed on 3 June 2023). This indicates that there are 3,298,912,062 bases in the human genome. When it is divided by 500,000 bp blocks, this yields 6598 such loci. We then used the hypergeometric calculator from the Graeber lab (https://systems.crump.ucla.edu/hypergeometric/index.php, accessed on 3 June 2023) to calculate the fold overrepresentation of these genomic intervals.

### 4.3. Predicted Causal Polygenic Risk Gene Annotation, Network Construction and Overlap

#### 4.3.1. Calculating Locus-Level Overrepresentation between SLE and SjD

To build the list of predicted causal SjD risk genes, we applied the same approach that we previously used with modification [30]. Once the list of genomic loci was defined, we annotated this list of SjD-associated loci using the output form of the Locus2Gene (L2G) algorithm [31] from Open Targets Genetics [32] as previously described. However, two loci were clearly misannotated using this approach based on published data, NCF1 [81] and C4A, C4B [13]. Therefore, they were manually reassigned. In addition, one large GWAS study of Sjögren’s Disease, that of Khatri et al. [29], has not been incorporated into either of these data sources. Therefore, we manually reviewed the unannotated variants from this study. We then assigned a predicted causal gene for each locus. To do so, we used a combination of manual annotation from the literature and selected a proxy from the list of SLE loci. We then assigned the predicted causal gene as the likely causal gene for SjD. Details of proxy selection are in (Appendix A). Detailed instructions can be found in the Appendix A. These instructions include step-by-step instructions on navigating the EBI GWAS Catalog [76]. They also include links to: resources to interpret EBI GWAS Catalog data tables [82]; documentation for the L2G algorithm [83]; and documentation for the open targets genetics database [84].

#### 4.3.2. Building the Predicted Causal Gene Network for SjD

To build the predicted protein–protein interaction network of causal SjD risk genes, we used the STRING database version 11.5 (http://string-db.org, accessed on 22 May 2023) [77]. We entered the list of SjD-predicted causal genes, ensured correct identifier mapping and then exported this network to Cytoscape v3.10.0 for further annotation and analysis. Default settings were used to generate the predicted protein–protein interaction network. The predicted causal genes for SjD can be found in Appendix A. Detailed instructions can be found in the Appendix A. They include links to resources to interpret STRING networks [85] and various tutorials and guides for navigating and using Cytoscape [86,87,88,89,90,91].

#### 4.3.3. Calculating Overlap

As above (Section 4.2.3), the Graeber lab hypergeometric calculator was used to calculate the degree of overlap in terms of risk genes with the following parameters: 19 genes overlapping networks, 23 predicted causal SjD genes, 131 predicted causal SLE genes and 19,303 genes in the STRING v11.5 network for *Homo sapiens* (https://string-db.org/organism_overview.html, accessed on 22 May 2023). Detailed instructions can be found in the Appendix A and includes a link to the Graeber lab hypergeometric calculator [92].

### 4.4. Mendelian Randomization Analysis

#### 4.4.1. Overview of Mendelian Randomization and Application of Instrumental Variable Analysis to Genetic Studies

Mendelian randomization (MR) is a causal inference method that has been developed to estimate and test causal effects between phenotypes [93]. MR applies instrumental variable analysis techniques, using genetic variants as the instrumental variables. In the example of two-sample MR, causality between an exposure phenotype and an outcome phenotype is assessed by evaluating the effect of the variants associated with the exposure phenotype on the outcome phenotype. Mendel’s laws of segregation and independent assortment ensure that non-linked variants (that is, independent variants not in linkage disequilibrium with one another) are a source variation potentially impacting the outcome phenotype. Inheritance of independent variants is a source of random variation unrelated to confounding factors. Furthermore, germline genetic variants are not susceptible to issues with reverse causality [94,95]. Therefore, the effect of variants on the exposure and the outcome phenotypes can be used to determine causality of the exposure on the outcome if certain conditions are met (Figure 8).

The main three conditions of MR are those of (1) relevance, (2) independence, and (3) exclusion restriction [96,97]. The relevance condition requires that a variant needs to be robustly associated with the exposure of interest. When MR is applied to GWAS studies, this often results in limiting variants under examination to those that pass the commonly accepted threshold for genome-wide significance, *p* < 5.0 × 10^−8^. The independence condition holds that variants are not associated with confounding factors impacting the association between the exposure phenotype and the outcome phenotype. The exclusion restriction condition holds that each variant is associated with the outcome only through the exposure of interest. In sum, MR is a powerful method to determine causal relationships between phenotypes that rely on specific conditions to be met regarding the relationship of genetic variants, confounding factors, and exposure/outcome phenotypes.

The putative causal gene-level overlap between SjD and SLE suggests that both phenotypes are mediated by genetic factors that impact the same unmeasured, intermediate phenotype. We posit that persistent loss of tolerance to nuclear autoantigens is a likely intermediate. With this framework, which one of the instrumental variable conditions is most important to attend to in our study of whether SjD and SLE are causally related? We have ensured that condition 1 (relevance) is met by setting a threshold for robust association of variants. Condition 2 (independence) seems unlikely as this would invoke a confounding factor that impacts both the genotype at these variants and the outcome phenotype. In the Introduction, we briefly reviewed the extensive body of literature supporting shared phenotypic, clinical, and molecular features of SjD and SLE. Considering these data, violation of the independence condition, seems less plausible. If independence were violated, a genetic risk factor impacts a confounding phenotype, C, that in turn impacts both SjD and SLE byinvoking them. While the extensive phenotypic sharing makes violation of the independence condition somewhat less likely, correlated pleiotropy remains possible in any MR analysis evaluating causal relationships between phenotypes [98]. Thus, of these three conditions, the most relevant to our study is that of exclusion restriction. When this condition is violated, the genetic variant impacts the outcome phenotype independently of the exposure phenotype or the unmeasured phenotype that causes it.

Importantly, there are several MR analysis methods that relax the exclusion restriction condition/assumption [93,97]. The basic approach to MR is similar to inverse-variance weighted meta-analysis. It involves weighting the ratio of the effect size in the exposure and the outcome by their associated uncertainty (inverse-variance weighted). This approach is valid under the assumption of balanced pleiotropy, where pleiotropic effects are equally as likely to be positive as negative. One method that allows for balanced pleiotropy in summary statistic MR analyses is the weighted median approach that selects and removes outliers. This approach assumes that the majority of the variants tested are valid [99]. Another approach, weighted mode, also selects and removes outliers. This approach remains valid even if a majority of the variants are valid but is overly conservative and sensitive to the addition/removal of variants [100]. Importantly, the weighted mode allows for directional pleiotropy. Another method that allows for directional pleiotropy is MR Egger, which also puts constraints on the pleiotropy. MR Egger assumes that the magnitude of pleiotropy across all variants is unrelated to the strength of association between the variant and the phenotype of interest. This assumption is termed the Instrument Strength Independent of Direct Effect or InSIDE assumption. The InSIDE assumption requires that the association between the genetic variant and the exposure is not correlated with a path from the variant to the outcome that is independent of the exposure of interest. When this assumption is violated and unbalanced pleiotropy is present, this will bias the MR Egger estimate of causal effect. This will occur when genetic variants influence the outcome via pleiotropy or through confounder impact on both the exposure and outcome [41,101]. While the outlier selection and removal methods attempt to limit the degree of heterogeneity between variants, the premise of MR Egger is for a set of relevant variants; the causal effects average out to a reasonable estimate of the true causal effect between the exposure and outcome phenotypes [41]. We refer the interested reader to five recent publications further explaining these methods [93,96,97,99].

#### 4.4.2. Evaluating the Causal Relationship between SjD and SLE

To evaluate if SjD and SLE are causally related through individual causal risk variants, we performed Mendelian randomization analysis. Because each of the four methods discussed above (Section 4.2.1) has different sensitivity and conditions, we applied all of them separately. With genetic variant data, a common mode of violation of instrumental variable conditions/assumptions is that the variant in question is not causal but is in linkage disequilibrium with a causal variant. The fine-scale linkage disequilibrium patterns differ between continental ancestral populations [49]. Therefore, wherever possible, we restricted our initial analyses to comparisons between SjD and SLE GWAS studies carried out within the same continental ancestral population. Specifically, we performed trans-ancestral MR in Analysis 4. Several GWAS of SjD have been carried out in cohorts of persons with East Asian ancestry. The most significant findings were reported in the respective publications, and we were able to use these data to select instrumental variables for Analysis 3. However, summary statistics for these studies are not currently available in a harmonized form in the 2-sample MR R library/MRC IEU open GWAS catalog. Thus, given this limitation, we performed trans-ancestral MR in Analysis 4. Our analysis scheme is shown (Figure 7).

#### 4.4.3. Selection of Instrumental Variables for SLE and SjD

We used lists of variants associated with SjD or SLE by GWAS. For SjD, these variants and links to the GWAS studies are listed in (Appendix A). For SLE, we used harmonized GWAS data sets from Bentham et al. [40] and Wang et al. [47] as well as from FinnGen [44]. All of these were accessed through the MR-Base web app v1.4.3 [43,102] interface for the ‘TwoSampleMR’ v0.5.5 R package [43,103]. A full listing of the GWAS studies used for the putative causal gene network and mendelian randomization analyses is listed below (Table 7).

In terms of summary statistics or the primary study, there was a great deal of variability. Some studies reported odds ratio and 95% confidence interval. Some studies reported the natural log of the odds ratio and its standard error. Other studies did not consistently indicate the effect allele for the odds ratio. We used the following equations to convert between odds ratio and beta/effect size scale.
Z = β/se(1)

Z is the Z score of a normal distribution, β is the natural logarithm of the odds ratio, and ln(OR) and se is the standard error of β.
[lower 95% CI of OR-upper 95% CI of OR] = [e^(β−1.96 se)^ − e^(β+1.96 se)^](2)
where the upper and lower CI are the limits of the 95% confidence interval of the OR and e is Euler’s number, the sum of the reciprocals of n! for all numbers, n, between 0 and infinity.
P = 2 ɸ(|−Z|)(3)
where P is the *p*-value and ɸ is the cumulative distribution function for a standard normal distribution.

#### 4.4.4. Quality Control of Instrumental Variables for SLE and SjD

One of the three assumptions of Mendelian randomization is that the instrumental variables are robustly associated with the exposure. To ensure this was the case for SjD, we used the conservative cutoff for genetic association at the genome-wide significance threshold of *p* < 5.0 × 10^−8^ in any single study (Appendix A). We sought to ensure that this was the case for the instrumental variables from SLE GWAS studies through MRBase. To do so, we visually inspected the regional association plots from GWAS in LocusZoom (https://my.locuszoom.org/, accessed on 30 May 2023) [109]. There were numerous variants that were robustly associated with SLE in Bentham et al. GWAS summary statistics [40] but looked suspicious for artefacts. These variants are in strong linkage disequilibrium (according to 1000 genomes project data) with one or more nearby variants that do not exhibit robust evidence for association. Review of the primary publication [40] revealed that these association signals were not replicated. Therefore, these variants were removed from our list of instrumental variables, as we deemed them to likely violate the relevance condition. Importantly, this problem of quality control has also been noted for Bentham et al. GWAS summary statistics available on dbGAP [110]. We used the same procedure to ensure that Wang et al. GWAS summary statistics [47] available in the EBI GWAS catalog did not harbor similar potential artifacts. We did not find any variants of this nature with this procedure.

#### 4.4.5. General Analysis Approach

Fine-scale linkage disequilibrium patterns differ between continental ancestral populations [49]. If the associated variants are not in linkage disequilibrium in a different ancestral population, this could represent a violation of the MR instrumental variable assumptions. So, where data were available, we compared variants within the ancestral population where the original GWAS was performed. This allows for the possibility that the variants identified are not bona fide disease causal variants but are in linkage disequilibrium with bona fide disease causal variants.

#### 4.4.6. MR Analysis Parameters

Linkage disequilibrium between variants is one of the ways in which MR conditions/assumptions can be violated for some MR analysis methods. Therefore, for each MR analysis presented, we used LD-based clumping to prune variants in LD with one another. Since our comparisons were against variants where we were confident regarding strand and effect allele as detailed in the Appendix A, we did allow for palindromic LD-based proxies to be used. We set a minor allele frequency cutoff for assigning palindromic variants of 0.3. The GWAS studies in ‘TwoSampleMR’ and MRBase.org underwent automated harmonization to a particular strand of the reference genome. Since the inaccurate assignment of palindromic variants could change the direction of an MR result, we excluded palindromic variants. For each analysis, we used the four MR analysis methods described: inverse-variance weighted, weighted median, weighted mode and MR Egger.

## 5. Conclusions

Our complementary analyses revealed pervasive sharing of causal genetic risk factors between Systemic Lupus Erythematosus and Sjögren’s Disease. This shared causal genetic risk may partly explain the clinical, immunologic, and molecular overlap between these diseases. Our study has implications for future genetic studies. Jointly modeling these two disease phenotypes as a single phenotype will likely increase power to detect shared and divergent risk factors. Likewise, there are also implications for clinical care that include the differential diagnosis of these two disease phenotypes and therapeutic selection where features of more than one disease are present.

## Figures and Tables

**Figure 1 ijms-24-14449-f001:**
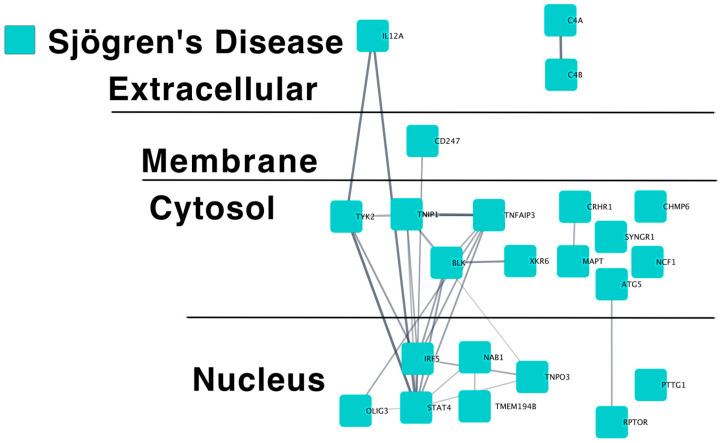
Sjögren’s Disease predicted causal genes for a distinct biological network. The predicted protein–protein interaction network from the STRING database of Sjögren’s Disease GWAS-identified putative causal genes is shown. (https://www.ndexbio.org/viewer/networks/99b77fa6-478b-11ee-aa50-005056ae23aa, accessed on 3 June 2023).

**Figure 2 ijms-24-14449-f002:**
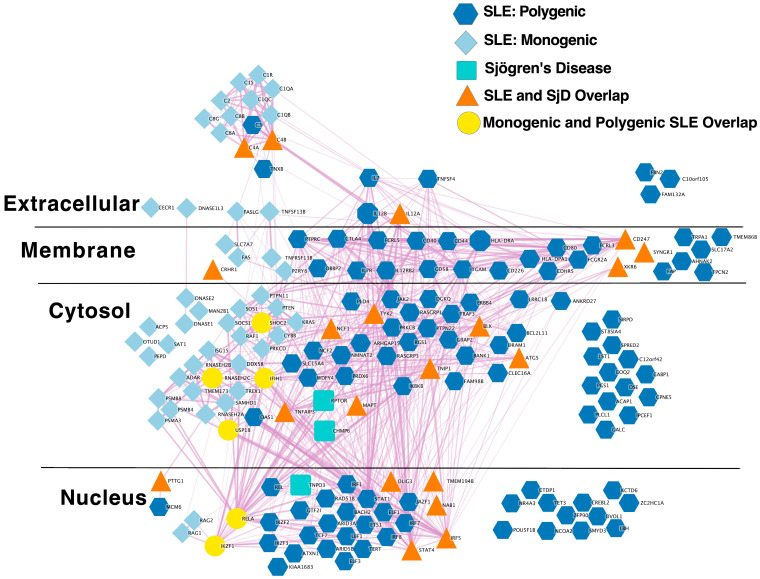
The Sjögren’s Disease putative causal genes network exhibits substantial overlap with that of Systemic Lupus Erythematosus. The predicted protein–protein interaction network from the STRING database of Sjögren’s disease GWAS-identified putative causal genes (teal rounded rectangles) is shown together with those of the polygenic human SLE (dark blue hexagons) and monogenic human SLE (light blue diamonds). Overlaps of SjD with either of the SLE networks are indicated with orange triangles and take priority over overlaps between monogenic and polygenic SLE (yellow circles).

**Figure 3 ijms-24-14449-f003:**
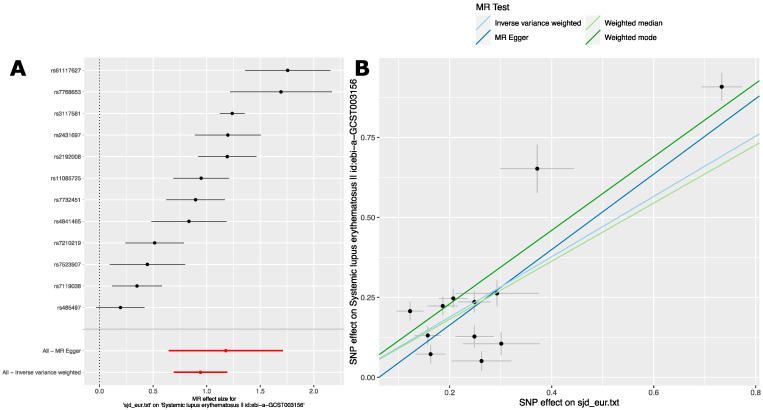
MR analysis reveals a robust and stable causal relationship between Sjögren’s Disease and Systemic Lupus Erythematosus in a GWAS of persons with European ancestry. (**A**) Forest plot of SjD effect estimate on SLE in the Bentham J. et al., GWAS [40] for individual risk variants. MR Egger and inverse-variance weighted overall estimates are also shown. Data are MR effect sizes (beta ± se). Black dots and black lines indicate MR effect estimates and confidence intervals for individual variants. Red dots and red lines indicate aggregate MR effect estimates and confidence intervals as indicated for MR Egger and Inverse variance weighted analyses. (**B**) scatter plot of MR effect sizes (beta) ± standard error (se).

**Figure 4 ijms-24-14449-f004:**
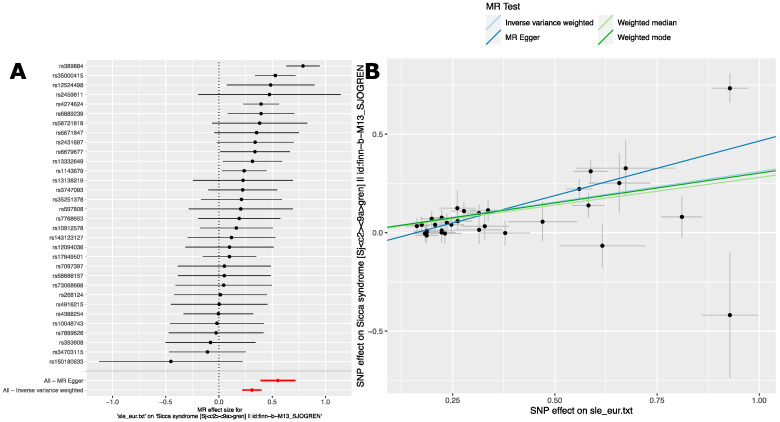
MR analysis reveals a robust and stable causal relationship between Systemic Lupus Erythematosus and Sjögren’s Disease in a GWAS of persons with European ancestry. (**A**) Forest plot of effect estimate of SLE on SjD (M13_Sjogrens) in the FinnGen GWAS [44] for individual risk variants. MR Egger and inverse-variance weighted overall estimates are also shown. Data shown are MR effect sizes (beta ± se). Black dots and black lines indicate MR effect estimates and confidence intervals for individual variants. Red dots and red lines indicate aggregate MR effect estimates and confidence intervals as indicated for MR Egger and Inverse variance weighted analyses. (**B**) scatter plot of MR effect sizes (beta) ± standard error (se).

**Figure 5 ijms-24-14449-f005:**
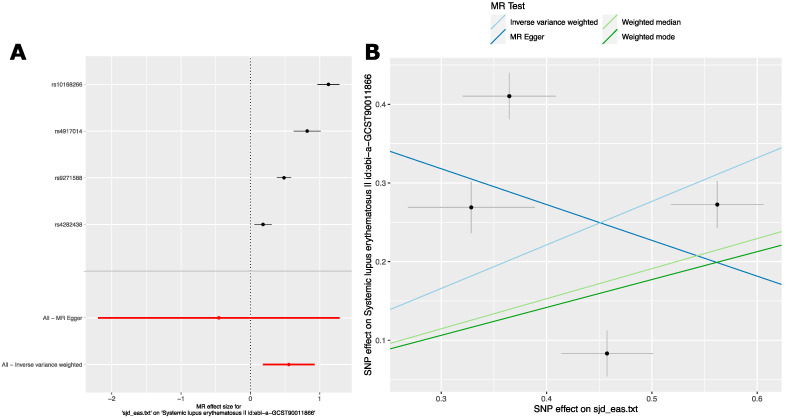
MR analysis reveals a causal relationship or pleiotropy between Sjögren’s Disease and Systemic Lupus Erythematosus in a GWAS of persons with East Asian ancestry. (**A**) Forest plot of SjD on SLE in the Wang, Y.F. et al., GWAS [47] for individual risk variants. MR Egger and Inverse-variance weighted overall estimates are also shown. Data shown are MR effect sizes (beta ± se). Black dots and black lines indicate MR effect estimates and confidence intervals for individual variants. Red dots and red lines indicate aggregate MR effect estimates and confidence intervals as indicated for MR Egger and Inverse variance weighted analyses. (**B**) scatter plot of MR effect sizes (beta) ± standard error (se).

**Figure 6 ijms-24-14449-f006:**
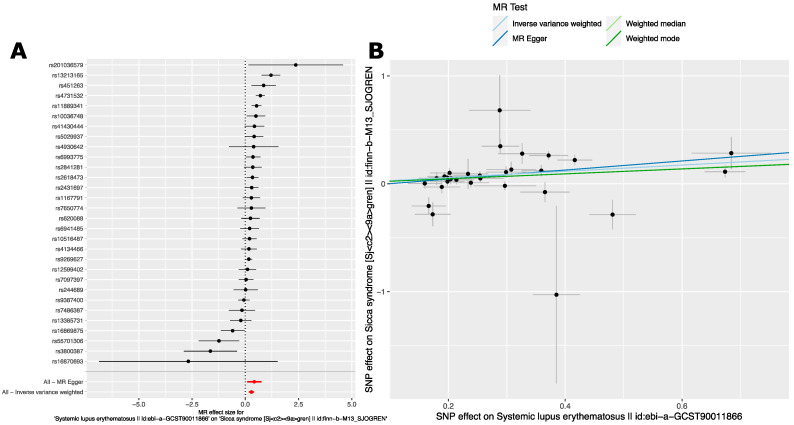
MR analysis reveals a causal relationship between Systemic Lupus Erythematosus in persons of East Asian ancestry and Sjögren’s Disease in GWAS of persons with European ancestry. (**A**) Forest plot of SLE from the Wang, Y.F. et al. GWAS [47] on SjD in FinnGen [44] for individual risk variants. MR Egger and Inverse-variance weighted overall estimates are also shown. Data shown are MR effect sizes (beta ± se). Black dots and black lines indicate MR effect estimates and confidence intervals for individual variants. Red dots and red lines indicate aggregate MR effect estimates and confidence intervals as indicated for MR Egger and Inverse variance weighted analyses. (**B**) scatter plot of MR effect sizes (beta) ± standard error (se) across the two conditions/populations.

**Figure 7 ijms-24-14449-f007:**
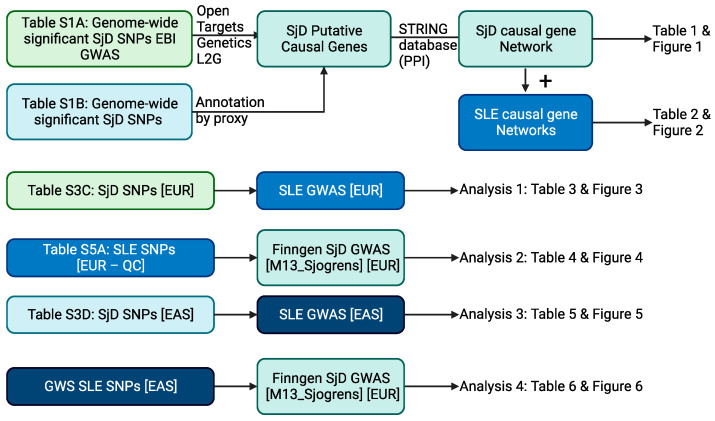
Analysis scheme. [EUR] indicates European ancestry GWAS. [EAS] indicates East Asian ancestry GWAS. M13_Sjogren’s indicates the phenotype code from FinnGen used to define Sjögren’s for that GWAS. [EUR–QC] indicates genome-wide significant variants after removal of variants likely violating the relevance condition as described in the methods. Created with BioRender.com (accessed on 6 June 2023).

**Figure 8 ijms-24-14449-f008:**
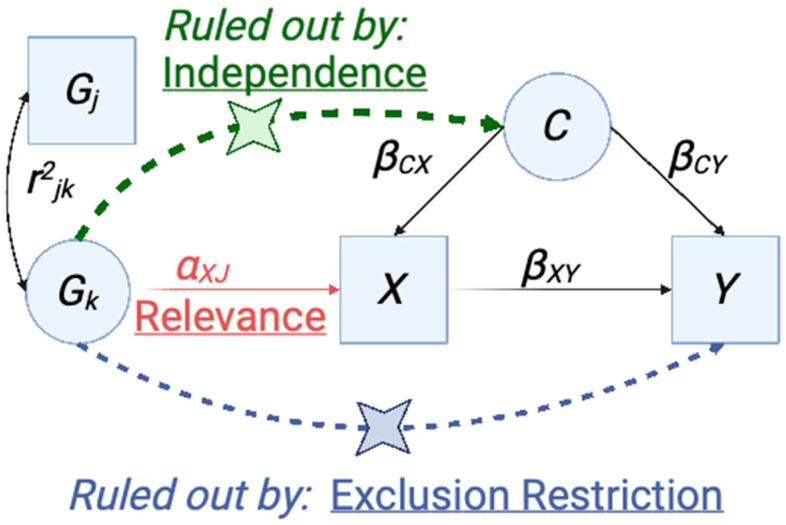
Mendelian randomization causal graph and conditions. In our study, the most likely scenario is that the chosen variant, *G_k_*, is in linkage disequilibrium with a causal variant, *G_j_*. In the event that this LD is perfect (r^2^ = 1) and the conditions of MR are met, then the association between *G_j_* and the outcome is fully mediated by the exposure. Therefore, the ratio of marginal effects of *G_j_* on X and *G_j_* on Y is equal to *β_XY_*. Effects of the variant *G_j_* through any confounder, *C*, or on *Y* are both ruled out through the Independence and Exclusion Restriction conditions. The relevance condition is that *Gj* (through *G_k_* in this case) is truly associated with the effect, *X*. In situations of comparison across populations with different fine-scale LD patterns, the relevance assumption is likely to not be met. Created with BioRender.com (accessed on 6 June 2023).

**Table 1 ijms-24-14449-t001:** Predicted causal gene network characteristics and explanations from the STRING user documentation [https://version11.string-db.org/help/getting_started/, accessed on 22 May 2023].

Network ^1^	Nodes ^2^	Edges ^3^	Degree ^4^	Clustering ^5^	Exp. Edges ^6^	*p* ^7^
Monogenic SLE	54	169	6	0.65	33	1.0 × 10^−16^
Polygenic SLE	127	497	8	0.44	107	1.0 × 10^−16^
Sjögren’s Disease	23	29	3	0.68	4	5.5 × 10^−16^

^1^ Putative causal gene network. ^2^ Number of nodes in the network. ^3^ Number of edges in the network. ^4^ The average node degree is a number of how many interactions (at the score threshold) that a protein has on average in the network. ^5^ The expected number of edges indicates how many edges is to be expected if the nodes were to be selected at random. The clustering coefficient is number between 0 and 1 that measures how connected the nodes in the network are. Highly connected networks have high values. ^6^ The expected number of edges indicates how many edges is to be expected if the nodes were to be selected at random. ^7^ A small PPI enrichment *p*-value indicates that the nodes are not random and that the observed number of edges is significant.

**Table 2 ijms-24-14449-t002:** Overlaps of SLE putative causal gene networks and GWAS loci with SjD.

Network ^1^	Exp. Overlap ^2^	Fold Enriched ^3^	*p* ^4^
Monogenic SLE	0.06	46	3.6 × 10^−05^
Polygenic SLE	0.15	130	1.1 × 10^−41^
All SLE	0.21	96	7.6 × 10^−39^
Locus Overlap	0.63	28	1.1 × 10^−24^

^1^ Either putative causal gene network or genetic risk loci. ^2^ The number of expected overlapping nodes assuming similar length lists were randomly selected from the genome (unassociated). ^3^ The fold over-represented compared to expectation. ^4^
*p*-value for hypergeometric distribution.

**Table 3 ijms-24-14449-t003:** MR results with SjD as exposure and SLE as outcome reveals a causal relationship in persons with European Ancestry.

Method ^1^	#SNPs ^2^	β ^3^	se ^4^	*p* ^5^
Inverse-variance weight	12	0.94	0.13	1.4 × 10^−13^
Weighted median	12	0.91	0.11	2.4 × 10^−15^
Weighted mode	12	1.15	0.12	9.7 × 10^−07^
MR Egger	12	1.18	0.27	0.0014

^1^ MR analysis method. ^2^ Number of variants used in the analysis. ^3^ MR effect size, beta. ^4^ standard error of beta. ^5^
*p*-value for causal association.

**Table 4 ijms-24-14449-t004:** MR results with SLE as exposure and SjD as outcome reveals causal relationship in persons with European Ancestry.

Method ^1^	#SNPs ^2^	β ^3^	se ^4^	*p* ^5^
Inverse-variance weight	32	0.31	0.04	1.6 × 10^−11^
Weighted median	32	0.28	0.05	2.4 × 10^−08^
Weighted mode	32	0.30	0.10	0.0034
MR Egger	32	0.554	0.08	2.7 × 10^−07^

^1^ MR analysis method. ^2^ Number of variants used in the analysis. ^3^ MR effect size, beta. ^4^ standard error of beta. ^5^
*p*-value for causal association.

**Table 5 ijms-24-14449-t005:** MR results with SjD as exposure and SLE as outcome reveals either a causal relationship in persons with East Asian Ancestry or pleiotropy.

Method ^1^	#SNPs ^2^	β ^3^	se ^4^	*p* ^5^
Inverse-variance weight	4	0.55	0.19	0.004
Weighted median	4	0.38	0.05	2.5 × 10^−12^
Weighted mode	4	0.35	0.05	0.003
MR Egger	4	−0.45	0.89	0.911

^1^ MR analysis method. ^2^ Number of variants used in the analysis. ^3^ MR effect size, beta. ^4^ standard error of beta. ^5^
*p*-value for causal association.

**Table 6 ijms-24-14449-t006:** MR results with SLE in persons with East Asian Ancestry as exposure and SjD in persons with European ancestry as outcome reveals a causal relationship.

Method ^1^	#SNPs ^2^	β ^3^	se ^4^	*p* ^5^
Inverse-variance weight	30	0.29	0.06	3.5 × 10^−05^
Weighted median	30	0.23	0.06	5.6 × 10^−06^
Weighted mode	30	0.23	0.08	0.006
MR Egger	30	0.42	0.17	0.019

^1^ MR analysis method. ^2^ Number of variants used in the analysis. ^3^ MR effect size, beta. ^4^ standard error of beta. ^5^
*p*-value for causal association.

**Table 7 ijms-24-14449-t007:** GWAS or other studies used in analysis or referenced in Appendix A.

1st Author [Ref] ^1^	*N_case_:N_ctl_ ^2^*	PMID ^3^	Other IDs ^4^	Note ^5^
Lessard [104]	1638:6754	24097067		EUR SjD GWAS
Li [105]	1845:3757	24097066		EAS SjD GWAS
Khatri [29]	3851:23652	35896530	phs002723.v1.p1	EUR SjD GWAS
Song [106]	420:15876	27503288		EAS SjD GWAS
Qu [107]	665:863	28552951		EAS SjD GWAS
Kurki [44]	2735:416383	36653562	M13_Sjogren finn-b-M13_SJOGREN	EUR SjD GWAS URL in ST3A
Sakaue [108]	303:175599	34594039	Sjogren_Syndrome	EAS SjD GWAS URL in ST3A
Bentham [40]	5201:9066	26502338	GCST003156 ebi-a-GCST003156	EUR SLE GWAS
Wang [47]	4222:8431	33536424	GCST90011866 ebi-a-GCST90011866	EAS SLE GWAS

^1^ 1st Author [Ref]—Name of first author as well as reference within this report. ^2^ Ncase:Nctl, number of persons with disease (cases) and persons without disease or population controls, separated by “:”. ^3^ The pubmed identifier for the primary publication. ^4^ Other study identifiers from dbGAP, Finngen, Biobank Japan, or the EBI GWAS Catalog. ^5^ Notes on the individual study.

## Data Availability

Public data from the original studies analyzed in this manuscript can be found in the original publications or via the identifiers listed in (Table 7) and (Appendix A). Further details of the analyses that we presented, including supporting logs, analysis R and python scripts and output files can be found at (https://github.com/harleyi/sjd_sle_overlap, accessed on 24 May 2023). Any pertinent information that may have been omitted will be furnished by the corresponding author upon reasonable request.

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
