# Peer review of "Pervasive Sharing of Causal Genetic Risk Factors Contributes to Clinical and Molecular Overlap between Sjögren’s Disease and Systemic Lupus Erythematosus"

_ijms, 2023, doi:10.3390/ijms241914449_

Round 1

Reviewer 1 Report

The article subject is interesting and the results well presented.

I would have some minor comments.

The introduction is too long, a maximum of 500words is generally recommended. 

The abbreviations should be explained only once, when first used.\

There are some limitations discussed. Even so, generally a distinct paragraph for the study limitations is welcomed.

Also, the novelty as this study could be noted at the end of introduction.

Author Response

We would like to thank Reviewer #1 for a careful and thoughtful review. We have revised the manuscript in response to the comments from Reviewer #1. We believe that these changes have improved our work.

The main changes to the manuscript in response to comments include:

  1. reduced the length of the introduction (Reviewer #1)
  2. moved sections of results to methods (Reviewer #2)
  3. included dedicated paragraph on study limitations in the discussion (Reviewer #1 & #2)
  4. made general revisions to improve conciseness and clarity (Reviewer #1 & #2)

We outline our response to each comment made by Reviewer #1 below:

Point 1: The article subject is interesting and the results well presented.

Thank you.

Point 2: I would have some minor comments.

The introduction is too long, a maximum of 500words is generally recommended.

We have moved several portions of the introduction to the discussion section. The word count for the introduction is now close to 500 words.

Point 3:The abbreviations should be explained only once, when first used.

We have reviewed the manuscript and revised to remove redundant abbreviations.

Point 4: There are some limitations discussed. Even so, generally a distinct paragraph for the study limitations is welcomed.

We have extended the discussion of the limitations and address the limitations in a distinct paragraph in the discussion.

Point 5: Also, the novelty as this study could be noted at the end of introduction.

We added an explanation detailing the novelty of this study at the end of the introduction.

Reviewer 2 Report

I think this is a very interesting paper. However, the paper is lengthy and could be presented in a more systematic manner.

As an example, much of the text in the results sections should be in the discussion or in the methods sections. I would consider making the manuscript more concise by cutting the word count down considerably or moving some of the text to the supplementary section.

I would also considering including a paragraph on your study limitations.

Finally, was there any patient involvement in the design or production of the manuscript? If so I would include a small statement to this effect.

English language is generally good but could do with a proof read.

Author Response

We would like to thank Reviewer #2 for a careful and thoughtful review. We have revised the manuscript in response to the comments from Reviewer #2. We believe that these changes have improved our work.

The main changes to the manuscript in response to comments include:

  1. reduced the length of the introduction (Reviewer #1)
  2. moved sections of results to methods (Reviewer #2)
  3. included dedicated paragraph on study limitations in the discussion (Reviewer #1 & #2)
  4. made general revisions to improve conciseness and clarity (Reviewer #1 & #2)

We outline our response to each comment made by Reviewer #2 below:

Point 1: I think this is a very interesting paper.

Thank you.

Point 2: However, the paper is lengthy and could be presented in a more systematic manner.

As an example, much of the text in the results sections should be in the discussion or in the methods sections. I would consider making the manuscript more concise by cutting the word count down considerably or moving some of the text to the supplementary section.

We have revised the manuscript to increase clarity and conciseness, by moving the overview of mendelian randomization and the analysis scheme to the methods section.

Regarding the comment that: "much of the text in the results sections should be in the discussion or in the methods sections."

We have moved text from the results section into the discussion.

We also considered moving the explanation of Mendelian Randomization to the supplement. However, we did not make this change, leaving it as an extended explanation in the methods. In our view, as a newer technique, understanding of Mendelian Randomization and its application is not widespread. Since this special issue is aimed at Molecular Pathogenesis of Sjogren's, we did not assume that most readers will be familiar with this technique to the extent that practitioners and analysts are. For this reason, we believe this section belongs within the manuscript and not in a supplemental note.

Point 3: I would also considering including a paragraph on your study limitations.

With the revisions requested above, there is now a distinct paragraph in the discussion detailing our study limitations.

Point 4: Finally, was there any patient involvement in the design or production of the manuscript? If so I would include a small statement to this effect.

No.

Reviewer 3 Report

In this article, Chau et al evaluated whether Sjögren’s disease (SjD) and Systemic Lupus Erythematosus (SLE) share causal genetic risk factors. They compared the causal genetic risk for SLE and SjD using three complementary approaches. 1) published GWAS results for these two diseases by analyzing the predicted causal gene protein-protein interaction networks of both diseases, 2) overlapping risk intervals, and 3) two-sample Mendelian randomization (two sample MR) using GWAS summary statistics to determine whether risk variants for SLE are causal for SjD and vice versa. They found that both the putative causal genes and the genomic risk intervals for SLE and SjD overlap 28- and 130- times more than expected by chance. They concluded that overlapping causal genetic risk factors were found for both diseases using complementary approaches. These observations support the hypothesis that shared genetic factors drive the clinical and pathobiologic overlap between these diseases. It is more accurate because it uses a large number of existing GWAS data. The paper also provides a more comprehensive understanding of the disease as a whole. Overall, it is well written. Some minor revisions are needed, as described below.

Minor concerns)

1) In line 772, you wrote “Funding: Please add.“ Please remove "Please add.

2) In line 778, “Informed Consent Statement: Informed consent was obtained from all subjects involved in the study.” Were there any patients who needed a consent form for this study? I may have missed it, but it was my understanding that data was not obtained from new individual patients for the database-based study.

Author Response

We would like to thank Reviewer #3 for a careful and thoughtful review. We have revised the manuscript in response to the comments from Reviewer #3. We believe that these changes have improved our work.

The main changes to the manuscript in response to comments include:

  1. reduced the length of the introduction (Reviewer #1)
  2. moved sections of results to methods (Reviewer #2)
  3. included dedicated paragraph on study limitations in the discussion (Reviewer #1 & #2)
  4. made general revisions to improve conciseness and clarity (Reviewer #1 & #2)

We outline our response to each comment made by Reviewer #3 below:

Point 1: Minor concerns)

1) In line 772, you wrote “Funding: Please add.“ Please remove "Please add.

We have removed these words.

Point 2: 2) In line 778, “Informed Consent Statement: Informed consent was obtained from all subjects involved in the study.” Were there any patients who needed a consent form for this study? I may have missed it, but it was my understanding that data was not obtained from new individual patients for the database-based study.

The understanding of Reviewer #3 is correct. The manuscript template from the publisher included this statement. We left it in place in the submitted manuscript to indicate that informed consent was obtained from all participants in the initial studies. Clearly leaving this statement in place is confusing. Since we accessed these data from databases as indicated in Table #7 and Table #S2, it has now been removed.